# Synthesis of Aliphatic Polycarbonates from Diphenyl Carbonate and Diols over Zinc (II) Acetylacetonate

**DOI:** 10.3390/molecules27248958

**Published:** 2022-12-16

**Authors:** Jun Feng, Jin Li, Jian Feng, Zhong Wei, Ziqing Wang, Xiaoling Song

**Affiliations:** 1Key Laboratory for Green Process of Chemical Engineering of Xinjiang Bingtuan, School of Chemistry and Chemical Engineering, Shihezi University, Shihezi 832003, China; 2Xinjiang Tianye (Group) Coporation Limited, Shihezi 832000, China

**Keywords:** Zinc (II) acetylacetonate, aliphatic polycarbonate, biphenyl carbonate, melt transesterification, Lewis acidity

## Abstract

APCs (aliphatic polycarbonates) are one of the most important types of biodegradable polymers and widely used in the fields of solid electrolyte, biological medicine and biodegradable plastics. Zinc-based catalysts have the advantages of being low cost, being non-toxic, having high activity, and having excellent environmental and biological compatibility. Zinc (II) acetylacetonate (Zn(Acac)_2_) was first reported as a highly effective catalyst for the melt transesterification of biphenyl carbonate with 1,4-butanediol to synthesize poly(1,4-butylene carbonate)(PBC). It was found that the weight-average molecular weight of PBC derived from Zn(Acac)_2_ could achieve 143,500 g/mol with a yield of 85.6% under suitable reaction conditions. The Lewis acidity and steric hindrance of Zn^2+^ could obviously affect the catalytic performance of Zn-based catalysts for this reaction. The main reasons for the Zn(Acac)_2_ catalyst displaying a higher yield and M_w_ than other zinc-based catalysts should be ascribed to the presence of the interaction between acetylacetone ligand and Zn^2+^, which can provide this melt transesterification reaction with the appropriate Lewis acidity as well as the steric hindrance.

## 1. Introduction

As an environment-friendly polymer, aliphatic polycarbonates (APCs) have attracted much attention in recent years for their widespread application in the fields of waterborne polyurethane, solid electrolytes and biological medicine; with a number-average molecular weight (M_w_) greater than 70 Kg/mol, they can also partly replace polyethylene as biodegradable plastics to solve the problem of white pollution [1,2,3,4]. Therefore, the effective synthesis of APCs with a high molecular weight has become more and more important. The current reported routes for APC synthesis mainly include copolymerization of CO_2_ with epoxides or diols [5,6], ring-opening polymerization of cyclic carbonates (ROP) [7] and condensation polymerization of dialkyl carbonate with aliphatic diols [8]. Moreover, both of the latter processes can also be considered as indirect utilization of CO_2_. Particularly, melt transesterification of dialkyl carbonate with aliphatic diols has established a bridge from CO_2_ to APCs, as dimethyl carbonate (DMC) and diphenyl carbonate (DPC) can be produced from CO_2_ and methanol or phenol [9].

Recently, the melt transesterification of dialkyl carbonate with diols to prepare high-molecular-weight polycarbonates has received increasing attention for synthesizing polymers with diverse structure and few catalysts. It is well known that the activity of a catalyst for this process is closely related to its acidity and chelating ability of metal specie. Therefore, numerous transition metal compounds were found to be highly active catalysts for this process, such as organotin oxides [10], titanium compounds [3,11] and even simple metal salts [12,13]. As highly effective polymerization catalysts, many zinc salts have been extensively used in the synthesis of various polymers due to their characteristics of being low cost, nontoxic and biofriendly [14,15,16]. In a previous paper, the Zn^2+^ cation was found to be able to activate the alcoholic hydroxyl groups by coordinating the oxygen atom in the hydroxyl group and inducing the transesterification reaction [13]. Pastusiak et al. [17] found that Zinc(II) acetylacetonate (Zn(Acac)_2_) could initiate the ring-opening polymerization of cyclic trimethylene carbonate resulting in high-molecular-weight poly(trimethylenecarbonate). Zn(Acac)_2_ was also found to be an excellent catalyst for polylactic acid synthesis [18].

Inspired by these referenced works, Zn(Acac)_2_ was first employed as a catalyst for the one-pot melt transesterification of DPC and aliphatic diols to synthesize high-molecular-weight APCs. Using ZnCl_2_ as a contrast, the reaction conditions were explored in detail with Zn(Acac)_2_ as a catalyst for its excellent catalytic performance. In addition, XPS was used to characterize the catalyst structure to understand the influencing factors of zinc salt in the melt transesterification reaction of DPC with diols.

## 2. Results and Discussions

### 2.1. Selection of Catalysts

The catalytic performance of various zinc compounds for the melt transesterification of DPC with BD to synthesize PBC was evaluated at 180 °C and 200 Pa with a reaction time of 90 min; the M_w_, Ð and corresponding yield are listed in Table 1. As reported in a previous paper [13], this reaction scarcely occurred in the absence of a catalyst due to the relatively low nucleophilicity of the hydroxyl group in BD, and no reaction fraction could be observed in the blank experiment. One can also see in Table 1 that the M_w_ of PBC obtained over Zn(Acac)_2_ is 69,400 g/mol at the given conditions, with a Ð and yield of 1.64 and 93.7%, respectively. As discussed in the literature [3,19,20], the presence of by-products, including tetrahydrofuran, cyclic carbonate and other volatile oligomers, would occur during the melt transesterification of dialkyl carbonates with BD, resulting in a decrease in the PBC yield. This reaction was also performed using other zinc compounds as catalysts; ZnO, ZnCO_3_ and Zn(OAc)_2_ were all found to demonstrate relatively less activity, producing PBCs with a M_w_ of 8300, 28,200 and 53,200 g/mol at the same conditions, respectively. Though the M_w_ of PBC derived with ZnCl_2_ possessed the maximum value of 82,300 g/mol, the PBC yield was the lowest among all the tested catalysts. Additionally, not all the zinc compounds were active in this reaction; for example, ZnSO_4_ was almost inert with no fraction and polymer detected under the same conditions.

Additionally, Ð seems to only depend on the M_w_ of PBC, and the higher the M_w_ value, the wider the Ð value. One can also see from Table 1 that increasing the molar ratio of Zn(Acac)_2_ to DPC from 0.025% to 0.1% increases the M_w_ of PBC from 9600 g/mol to 69,400 g/mol; continuing to increase the Zn(Acac)_2_ concentration to 0.2% led to an obvious decrease in the M_w_ and yield of PBC to 62,500 g/mol and 88.4%, respectively. Obviously, a molar ratio of Zn^2+^ to DPC of 0.1 mol% should be the suitable catalyst amount for this reaction considering the M_w_ and yield comprehensively. Therefore, Zn(Acac)_2_ was selected as the model catalyst for further research.

### 2.2. Characterization of Poly (Butylene Carbonate)

The resultant polymers obtained over Zn(Acac)_2_ and ZnCl_2_ were analyzed by FT-IR and ^1^H NMR, and the results are shown in Figure 1. As observed in Figure 1a, the FT-IR spectra of the two samples are very close to each other. The absorption bands appearing at 2963 cm^−1^ and 2875 cm^−1^ are attributed to the asymmetric and symmetric C-H stretching vibration of methylene, respectively. The strong absorption bands at 1744 cm^−1^ and 1249 cm^−1^ can be ascribed to the stretching and asymmetric stretching vibrations of C=O and O-C-O of the carbonate backbone, respectively [3,13]. One can also see that all the synthesized PBC samples described herein have identical ^1^H NMR spectra, and only two strong signals appearing at 4.12 and 1.73 ppm can be observed in Figure 1b for both samples, which are attributed to a and b protons from BD units. No remarkable feature for the end-group was detected with their chemical shift at 3.64 or 7.32–7.38 ppm, suggesting the resultant polymers bear rather high molecular weight. Additionally, no peak at 3.4–3.5 ppm can be observed in the ^1^H-NMR spectrum of the PBC polymer, indicating there is no ether linage (-CH_2_-O-CH_2_-) in the PBC polymer, which is not hydrolysable and decreases the mechanical properties of the polymer [3]. Obviously, all the peaks for the two polymers are well concordant with the standard spectrum of PBC, and it is consistent with what is expected for a PBC structure [3,11,12,13].

### 2.3. XPS of Catalyst

The electronic property of Zn^2+^ in ZnCl_2_ and Zn(Acac)_2_ was also further examined using XPS to understand the relationship between catalytic performance and the nature of the catalyst; the results are illustrated in Figure 2. One can see that the binding energy of Zn 2p_3/2_ in the Zn(Acac)_2_ catalyst appeared at 1021.7 eV, ascribed to the presence of bivalence of Zn(II) [21,22]. The binding energy of ZnCl_2_ appeared at 1022.8 eV, which was higher than that of Zn(Acac)_2_. That is to say, the Lewis acid strength of ZnCl_2_ is stronger than that of Zn(Acac)_2_ in view of the concept for Lewis acid.

### 2.4. Effect of Reaction Conditions

In order to obtain the optimum conditions and further understand the relationship between the structure and catalytic performance of the catalyst, the melt transesterification of DPC and BD was performed under various reaction parameters with the ZnCl_2_ catalytic system as the control experiment. The effect of polymerization temperature was first examined in the range of 160–210 °C. The results, shown in Figure 3a, indicate that the M_w_ values of PBC over Zn(Acac)_2_ and ZnCl_2_ sharply increased when raising the polymerization temperature from 160 to 190 °C, which can often be ascribed to the acceleration of the diffusion-limited polycondensation kinetics due to the decrease in polymer viscosity at a higher temperature [23]. Then, the M_w_ gradually decreased as the temperature continuously increased. As for the ZnCl_2_ catalyst, the optimum temperature for the highest M_w_ values of 102,400 g/mol can be observed at 190 °C, while that for Zn(Acac)_2_ is 200 °C. Clearly, the M_w_ for ZnCl_2_ at a lower temperature is much higher than that of Zn(Acac)_2_, indicating that strong Lewis acidity seems to be a positive factor for the improvement of polymerization rate. Simultaneously, it is well known that there would be a completion effect between polymerization and decomposition during the whole process because the melt transesterification of DPC and BD is a typical reverse reaction. Therefore, pure ZnCl_2_ more easily expedites the decomposition and depolymerization of the obtained PBC polymer, which often can be explained by the fact that strong Lewis acidity is prone to attack the carbonyl oxygen atoms in APCs and impede the growing of the polymer chain to lower the M_w_ and yield [24]. This is also in accordance with the results shown in Figure 3b, in which the yield for the ZnCl_2_ catalyst sharply decreased with the rise of temperature. The excellent catalytic performance of Zn(Acac)_2_ may be explained by the fact that the coordination bond formed between the acetylacetone ligand and Zn^2+^ not only can decrease the Lewis acidity of the central Zn^2+^ but can also make Zn^2+^ attack carbonyl oxygen atoms and hinder undesirable side reactions [19]. Hence, the connection of -OH and the -OC(O)OC_6_H_5_ end-group while removing the generated phenol at reduced pressure to increase the polymer molecular chain proceeded smoothly. Obviously, 200 °C should be selected as the suitable polymerization temperature for the Zn(Acac)_2_ catalyst considering its M_w_ and yield comprehensively.

Figure 4 shows the dependence of the M_w_ and yield of PBC *versus* reaction time over different catalysts. As shown in Figure 4a, under a short reaction time of 30 min, the M_w_ was only 67,400 g/mol over Zn(Acac)_2_. As the reaction proceeded, the M_w_ of PBC rapidly increased to 143,500 g/mol with increasing reaction time to 120 min. However, when the time was beyond 120 min, the value for M_w_ showed no significant improvement. Likewise, the M_w_ values for ZnCl_2_ also increased as the reaction time was prolonged, and the M_w_ reached the maximum value of 122,500 g/mol at 60 min; then, the M_w_ of PBC would rapidly decline to 12,000 g/mol with further increases of time to 150 min. The reason for this phenomenon might be that this polymerization process very easily proceeds at the beginning, but with an increase in molecular weight, the viscosity of the reaction system becomes high, which causes a negative influence to further polymerization [13,19]. Therefore, excessive reaction time could enhance its reverse reaction and lead to the decrease in M_w_. Meanwhile, one can also see in Figure 4b that the yield of PBC continuously decreases with the prolongation of reaction time, which can be reasoned by the presence of a side reaction and sublimation of oligomer as reported in previous works [20]. Considering the above results, a temperature of 200 °C and a reaction time of 120 min were selected as the optimum reaction conditions for realizing the highest M_w_ with satisfactory yield for Zn(Acac)_2_.

### 2.5. Catalytic Activity towards Other Diols

To evaluate the potential and general application range of the Zn(Acac)_2_ catalyst, the catalytic melt transesterification of DPC with a verity of aliphatic diols, including 1,3-propanediol (PPD), 1,5-pentanediol (PD) and 1,6-hexanediol (HD) to synthesize corresponding aliphatic polycarbonates, poly(trimethylene carbonate) (PTMC), poly(pentamethylene carbonate)(PPMC), poly(hexamethylene carbonate) (PHC) and poly(hexamethylene)-co-poly(butylene carbonate) (PHBC) were investigated. As shown in Table 2, other common aliphatic diols, including PD, HD and their mixtures can also undergo efficient transesterification with DPC to high-molecular-weight APCs with high yields under the optimized reaction conditions. The inferior catalytic performance of PPD was thought to be related to its low boiling point or the poor thermal stability of the corresponding PTMC polymer [13].

## 3. Experimental Section

### 3.1. Materials

Commercial DPC, purchased from Guanghua Scitech Co., Ltd., Shenzhen, China, was purified by recrystallization in absolute ethyl alcohol. 1,4-butanediol (BD, 98%) was dehydrated by distillation over calcium hydride under dry nitrogen gas. ZnCl_2_, Zn(OAc)_2_, Zn(NO_3_)_2_ and ZnSO_4_ were obtained from Chengdu Kelong Chemical Reagent Co., Chengdu, China, and were dehydrated as described in the literature [11] before use. Other chemicals and catalysts were used without any further purification and treatment.

### 3.2. Synthesis of APCs

PBC polymer was synthesized by a one-pot melt polymerization method [13]. Typically, DPC (21.41 g, 0.1 mol), BD (9.01 g, 0.1 mol) and a certain amount of catalyst were charged into a 150 mL three-necked flask equipped with a mechanical stirrer, reflux condenser and thermometer. The reaction mixture was heated to 120 °C under stirring for a certain time until it became homogeneous under a nitrogen atmosphere. Then, a lower pressure (*ca* 200 Pa) was applied slowly over a period of *ca* 20 min to carry out the melt transesterification reaction at the given temperature. After a certain time, the pressure was returned to atmospheric pressure, and the volatile by-products could be removed through the reflux condenser. The resulting PBC polymer was separated by dissolving in CH_2_Cl_2_ and precipitating with ethanol, and then dried under vacuum at 50 °C for 12 h.

### 3.3. Characterization

The chemical structures of the resulting PBCs were identified by ^1^H-NMR. The ^1^H-NMR spectra were acquired in CDCl_3_ at 25 °C with a Bruker DRX-300 NMR (Brucker, Romanshorn, Switzerland) spectrometer. The weight-average molecular weight (M_w_) and dispersity (Ð) of the obtained APCs were determined by gel permeation chromatography (GPC). The GPC measurements were carried out at 30 °C on a Waters 515 HPLC system (Waters, Milford, MA, USA) equipped with a 2690D separation module and a 2410 refractive index detector. Tetrahydrofuran (THF) was used as the eluent at a flow rate of 0.5 mL/min. Polystyrene with a narrow molecular weight distribution was used as the standard for calibration. All the given data in this study were collected by averaging the scores on at least three tests.

Fourier transform infrared spectroscopy (FT-IR) was carried out on a Nicolet-38 FT-IR spectrometer (Thermo Electron, Boston, MA, USA) in the range of 400–4000 cm^−1^ using the KBr pellet technique. The TGA experiments were carried out using a Q600 SDT thermal analysis machine (TA instrument, Waltham, MA, USA) under a flow of N_2_ in a temperature range from 50 °C to 550 °C with a heating rate of 10 °C/min. The binding energy values and the atomic surface concentration of the corresponding elements of the samples were analyzed by X-ray photoelectron spectroscopy and performed on an ESCA LAB 250 photoelectron spectroscope at 3.0 × 10^−10^ mbar with a hemispherical analyzer and monochromatic Mg Kα radiation (*E* = 1253.6 eV). All the binding energies were referenced to the C1s peak at 284.5 eV of the surface adventitious carbon.

## 4. Conclusions

The catalytic properties of zinc (II) acetylacetonate in melt transesterification of diphenyl carbonate with aliphatic diols were investigated. The M_w_, yield and Ð of the obtained PBCs are influenced by the catalyst used, reaction temperature and time. Lewis acidity is found to be dominant for the polymerization rate at lower temperature, and increasing steric hindrance seems to give a positive effect on the improvement of yield. Therefore, the rise in M_w_ and yield for Zn(Acac)_2_ compared with ZnCl_2_ is mainly due to the decrease in Lewis acidity and the increase in steric hindrance.

## Figures and Tables

**Figure 1 molecules-27-08958-f001:**
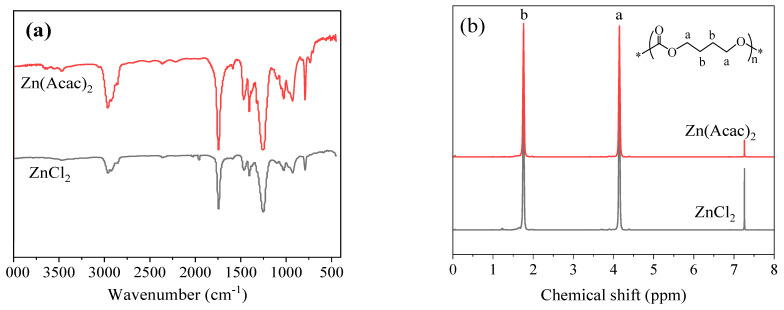
(**a**) FT-IR spectra and (**b**) ^1^H-NMR spectra of resultant copolymers obtained over ZnCl_2_ and Zn(Acac)_2_.

**Figure 2 molecules-27-08958-f002:**
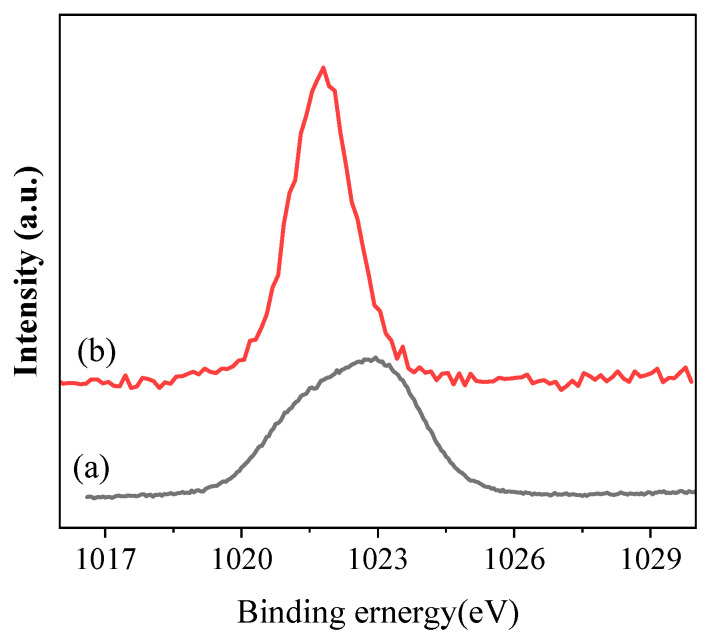
XPS spectra of Zn 2p_3/2_ for (**a**) ZnCl_2_ and (**b**) Zn(Acac)_2_.

**Figure 3 molecules-27-08958-f003:**
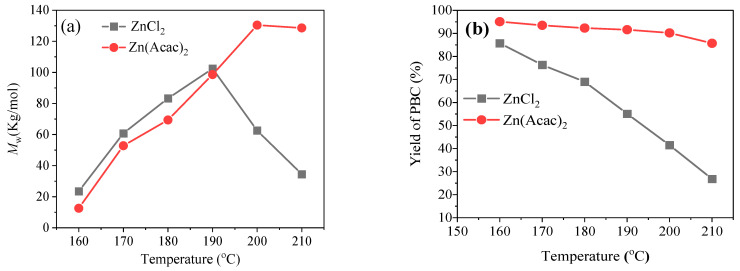
The effect of reaction temperature on the (**a**) M_w_ and (**b**) yield of PBC polymer over Zn(Acac)_2_ and ZnCl_2_. Reaction conditions: molar ratio of Zn^2+^ to DPC 0.10%, reaction time 90 min.

**Figure 4 molecules-27-08958-f004:**
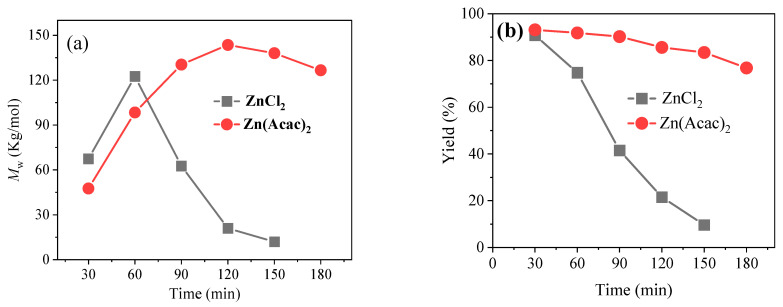
The effects of reaction time on the (**a**) M_w_ and (**b**) yield of PBC polymer over ZnCl_2_ and Zn(Acac)_2_. Reaction conditions: molar ratio of Zn^2+^ to DPC 0.1%, reaction temperature 200 °C.

**Table 1 molecules-27-08958-t001:** The catalytic performance of various zinc-based catalysts for the condensation polymerization of BD and DPC to PBC ^a^.

Entry	Catalyst	M_w_ (Kg/mol)	Ð (Mw/Mn)	PBC Yield ^b^ (%)
1	ZnCl_2_	82.3	1.72	69.0
2	Zn(NO_3_)_2_	13.2	1.63	93.2
3	ZnSO_4_	n.d	n.d	n.d
4	ZnCO_3_	28.2	1.57	93.5
5	ZnO	8.3	1.46	96.2
6	Zn(OAc)_2_	53.2	1.62	92.5
7	Zn(Acac)_2_	69.4	1.64	93.7
8 ^c^	Zn(Acac)_2_	9.6	1.52	95.2
9 ^d^	Zn(Acac)_2_	42.8	1.63	92.8
10 ^e^	Zn(Acac)_2_	62.5	1.64	88.4

^a^ Reaction conditions: reaction temperature 180 °C, reaction time 90 min, reaction pressure 200 Pa, the molar ratio of Zn^2+^ to DPC 0.1 mol %; ^b^ Yield expressed as a percentage of the theoretical value, which was calculated based on the 100% conversion of DPC to APCs; ^c^ the molar ratio of Zn^2+^ to DPC 0.025%; ^d^ the molar ratio of Zn^2+^ to DPC 0.05%; ^e^ the molar ratio of Zn^2+^ to DPC 0.2 mol%.

**Table 2 molecules-27-08958-t002:** Melt transesterification of DPC with a verity of diols in the presence of the Zn(Acac)_2_ catalyst ^a^.

Entry	Diols	Product	M_w_ (Kg/mol)	Yield (%)	Ð
1	PPD	PTMC	13.2	42.7	1.36
2	BD	PBC	143.5	85.6	1.76
3	PD	PPMC	141.5	93.6	1.81
4	HD	PHC	159.2	92.3	1.80
5	BD + HD ^b^	PBHC	152.4	89.6	1.80

^a^ Reaction conditions: reaction temperature 200 ℃, reaction time 120 min, the molar ratio of Zn^2+^ to DPC 0.1 mol %; ^b^ the molar ratio of BD to HD is 1:1.

## Data Availability

Not applicable.

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
