# Peer review of "Synthesis of Aliphatic Polycarbonates from Diphenyl Carbonate and Diols over Zinc (II) Acetylacetonate"

_molecules, 2022, doi:10.3390/molecules27248958_

Round 1
Reviewer 1 Report
Authors studied catalytic properties of zinc (II) acetylacetonate as an alternative for transesterification of di-phenyl carbonate with aliphatic diols and compared results with other zinc catalyst. Authors showed clearly that results are influenced by the type of catalyst usage, reaction temperature and time. Lewis acidity was demonstrated as dominant factor for the polymerization rate at lower temperature, and increasing steric hindrance was shown to provide a positive effect on yield improvement. The results can contribute to the literature especially for aliphatic polycarbonate synthesis and catalyst.
I recommend publication after minor revision.
The incomplete sentence at page 5 line 155 “bivalence 154 Zn(II)[xx]. The binding energy of ZnCl2 appeared at xx eV” should be corrected.
Writing mistakes such as Zn2P3 without subscript should be corrected.
Non English references should be corrected such as reactions[高分子学报] at page 6 line 185.
Error range or any other measure of the uncertainty should be given where required and possible.
Any impurities, side products or branching data should be reported if observed in characterization.
Basic first principle DFT calculations can support discussions for Zn (Acac)2 compared with ZnCl2 if authors have computational infrastructure or collaboration.
Author Response
Response to reviewer 1
Dear editor and reviewers,
Thank you very much for your valuable comments on our manuscript “Synthesis of aliphatic polycarbonates from diphenyl carbonate and diols over zinc (II) acetylacetonate” (Manuscript ID: Molecules-2061087) and give me so much useful comments to further perfect our manuscript. We have revised our original manuscript carefully based on the suggestions of reviewers and made the answers according to the reviewer’s questions point by point. The following is the answers to the referee’s points:
Reviewer #1
Authors studied catalytic properties of zinc (II) acetylacetonate as an alternative for transesterification of di-phenyl carbonate with aliphatic diols and compared results with other zinc catalyst. Authors showed clearly that results are influenced by the type of catalyst usage, reaction temperature and time. Lewis acidity was demonstrated as dominant factor for the polymerization rate at lower temperature, and increasing steric hindrance was shown to provide a positive effect on yield improvement. The results can contribute to the literature especially for aliphatic polycarbonate synthesis and catalyst. I recommend publication after minor revision.
(1) The incomplete sentence at page 5 line 155 “bivalence 154 Zn(II)[xx]. The binding energy of ZnCl2 appeared at xx eV” should be corrected.
A: Thanks very much for your so careful comments, and this sentence has been supplemented, and this sentence should be “One can see that the binding energy of Zn 2p3/2 in Zn(Acac)2 catalyst appeared at 1021.7 eV, ascribing to the presence of bivalence Zn(II)[20,21]. The binding energy of ZnCl2 appeared at 1022.8 eV”.
(2) Writing mistakes such as Zn2P3 without subscript should be corrected.
A: Thanks very much for your so careful comments, the error of“Zn2P3”has been corrected as “Zn 2p3/2”
(3) Non English references should be corrected such as reactions[高分子学报] at page 6 line 185.
A: Thanks very much for your so careful comments, this non-English reference has been corrected as literature [30] in the resubmitted revision.
(4) Error range or any other measure of the uncertainty should be given where required and possible.
A:Thank you very much for your useful instruction. It is well known reasonable error seems inevitable for all the experimental subjects. In fact, all the given data were collected by averaging the testing values of repeated measurements to lower the uncertainty. Now, the principle for data collection has been added in experimental section.
(5) Any impurities, side products or branching data should be reported if observed in characterization.
A:Thanks very much for your nice instruction, some side products including tetrahydrofuran, cyclic carbonate and other volatile oligomers would occur during the melt transesterification of biphenyl carbonate with 1,4-butanediol due to the presence of some side reactions. The specific reaction processes and the formation path of side products have been systematically discussed in previous works (Polymer Internatioanl,2011,60(7): 1060, Acta Polymerica Sinica, 2016, (12):1654 and European Polymer Journal, 2017,97:253). Therefore, the side products and branching data were not repeatedly reported in our work. According to the suggestion of reviewer, the information about side products has been added in this revised manuscript.
(6) Basic first principle DFT calculations can support discussions for Zn (Acac)2 compared with ZnCl2 if authors have computational infrastructure or collaboration.
A: Thanks very much for your nice suggestion, just as reviewer mentioned that first principle DFT calculations can further support discussions for Zn(Acac)2 compared with ZnCl2. However, as reported in previous works (Polymer International, 2011,60(7): 1060 and European Polymer Journal,2017,97:253), the reaction pathways to product and side products are very complex, the reaction mechanism and promotion effect of catalyst will be specially researched in the following works based on the DFT calculations.

Reviewer 2 Report
In this report, Jun et al. gave a systematic evaluation of the Zn(acac)2 promoted melt transesterification of diol and biphenyl carbonate. The investigation and characterizations are sound, which met the level of Polymers. Hence, I suggest its publication after the following issues are well addressed.
1. Page 3, table 1, Why Zn(acac)2 gave the best performance? Was it related to the different Lewis acidities on Zn2+ that was caused by the different anionic ligands (Cl-, NO3-, Acac, etc.)? or other reasons, such as the solubility of the Zinc catalysts?
2. Page 3, line 140, besides NMR, MALDI-TOF is another powerful strategy to detect the end groups. It can also give information whether linear or cyclic structures were formed during the reactions. So, it is strongly suggested to be carried out.
3. Page 5, line 153, please specify the term of “Zn2P3”,
4. Page 5, line 155, please provide the value in stead of “XX”,
5. During the reasons, was there any small molecules, such as cyclic or linear esters, were detected? So, GC-MS characterization for the reaction mixture is suggested.
6. Page 6, line 185, please cite the paper properly.
Other minor issues should also be carefully addressed:
1. Page 2, line 55, please specify the term of ‘DPC’ when it was firstly mentioned in the text.
2. Page 3, line 115, it should be “DPC from 0.025%”,
3. Page 3, line 115, the word “Continuing” should not be capitalized,
4. Page 3, line 117, it should be “62500 g/mol”,
5. Page 4, line 129, please use the standard description of HNMR in the caption of Figure 1.
6. Page 7, line 213, the word “1,3-propanediol” should not be capitalized,
7. Page 7, line 219, it should be “yields”,
8. Page 7, line 222, the term of “PVP” should be specified.
Author Response
Response to Reviewer 2
Dear editor and reviewers,
Thank you very much for your valuable comments on our manuscript “Synthesis of aliphatic polycarbonates from diphenyl carbonate and diols over zinc (II) acetylacetonate” (Manuscript ID: Molecules-2061087) and give me so much useful comments to further perfect our manuscript. We have revised our original manuscript carefully based on the suggestions of reviewers and made the answers according to the reviewer’s questions point by point. The following is the answers to the referee’s points:
Reviewer #2
In this report, Jun et al. gave a systematic evaluation of the Zn(acac)2 promoted melt transesterification of diol and biphenyl carbonate. The investigation and characterizations are sound, which met the level of Polymers. Hence, I suggest its publication after the following issues are well addressed.
(1) Page 3, table 1, Why Zn(acac)2 gave the best performance? Was it related to the different Lewis acidities on Zn2+ that was caused by the different anionic ligands (Cl-, NO3-, Acac, etc.)? or other reasons, such as the solubility of the Zinc catalysts?
A:Thanks very much for your instructive suggestions. As discussed in previous works, the transesterification reaction is a typical acid-catalyzed process, and the catalytic performance is mainly related to the acidity of zinc catalysts. As discussed in previous works (Acta Polymerica Sinica, 2016, (12):1654-1661), the improvement in acidic strength could enhance this reaction, while the increase of steric hindrance would reduce the side reactions, elevateing APCs product yield. So the raise in Mw and yield for Zn(Acac)2 compared with ZnCl2 is mainly resulted from the decrease of Lewis acidity and the increase of steric hindrance, this conclusion has been disclosed in the resubmitted version.
Furthermore, the zinc-based catalysts seem not soluble in the reaction materials, and the catalyst can be separated by the simple filtration of APCs dichloromethane solution (Chemical Research in Chinese Universities,32 (2016): 512 and RSC Advances,5 (2015) 87311).
(2) Page 3, line 140, besides NMR, MALDI-TOF is another powerful strategy to detect the end groups. It can also give information whether linear or cyclic structures were formed during the reactions. So, it is strongly suggested to be carried out.
A: Thanks very much for your nice instruction, just as reviewer mentioned some side products including tetrahydrofuran, cyclic carbonate and other volatile oligomers would occur during the melt transesterification of biphenyl carbonate with 1,4-butanediol. The specific reaction processes and the formation path of side products have been systematically discussed based on the results of MALDI-TOF in previous works (Polymer Internatioanl,2011,60(7): 1060, Acta Polymerica Sinica, 2016, (12):1654 and European Polymer Journal, 2017,97:253). Therefore, the side products were not repeatedly reported in this work. According to the suggestion of reviewer, the information about side products has been added in this revised manuscript.
(3) Page 5, line 153, please specify the term of “Zn2P3”,
A:We have corrected “Zn2P3” into “Zn 2p3/2” in the resubmitted manuscript.
(4) Page 5, line 155, please provide the value instead of “XX”,
A:This value has been added in the resubmitted version.
(5) During the reasons, was there any small molecules, such as cyclic or linear esters, were detected? So, GC-MS characterization for the reaction mixture is suggested.
A: Some small molecule side products have been widely reported in previous works during the melt transesterification of biphenyl carbonate with 1,4-butanediol to PBC (Polymer Internatioanl,2011,60(7): 1060, Acta Polymerica Sinica, 2016, (12):1654 and European Polymer Journal, 2017,97:253), including tetrahydrofuran, cyclic carbonates, and other volatile oligomers. In our work, the same small molecule side products were detected, therefore, the side products were not repeatedly discussed. According to the suggestion of reviewer, the information about side products has been added in this revised manuscript.
(6) Page 6, line 185, please cite the paper properly.
A: This literature has been cited in the revised manuscript.
(7) Other minor issues should also be carefully addressed:
7.1 Page 2, line 55, please specify the term of ‘DPC’ when it was firstly mentioned in the text.
A: The term of “DPC” has been specified for diphenyl carbonate in line 26 when it was firstly mentioned in the text.
7.2. Page 3, line 115, it should be “DPC from 0.025%”,
A:Thanks very much for so careful comments, and the mentioned mistake has been corrected as “DPC from 0.025%”.
7.3. Page 3, line 115, the word “Continuing” should not be capitalized,
A:The word “Continuing” has been changed into lowercase “continuing”.
7.4. Page 3, line 117, it should be “62500 g/mol”,
A:The unit of measurement of “g/mol” has been added into the revised version.
7.5. Page 4, line 129, please use the standard description of HNMR in the caption of Fig.1.
A: In the revised version, the standard description of“1H NMR”has been substituted for “HNMR” in the caption of Figure 1.
7.6 Page 7, line 213, the word “1,3-propanediol” should not be capitalized,
A:The word “1,3-Propanediol” has been changed into lowercase “1,3-propanediol”.
7.7. Page 7, line 219, it should be “yields”,
A:The word “yield” has been corrected as “yields” in the resubmitted version.
7.8. Page 7, line 222, the term of “PVP” should be specified.
A:This mistake has been corrected as “Zn(Acac)2” in the resubmitted version.

Reviewer 3 Report
The authors have explored the Synthesis of aliphatic polycarbonates from diphenyl carbonate and diols over zinc (II) acetylacetonate. Reaction optimization was done, and alittle substrate scope has been investigated.
The manuscript is poorly set up and the language of the manuscript is poor. consequently, extensive English editing is required.
LINE 55 (aliphatic diols to synthesis) should be to synthesize
LINE 64 (was purification by recrystallization) should be changed to (was purified by)
LINE 82 (Mw and PDI of the obtained) there is no definition for PDI throughout the manuscript. Authors should give a definition for its first appearance in the manuscript
LINE 100 (DPC with BD to synthesis) should be changed to ( to synthesize)
LINE 105 (Zn(Acac)2 is 69 400 g/mol) please correct the numbers (delete the space)
LINE 102, 103 (this reaction scarcely occurred in the absence of a catalyst due to the relatively low nucleophilicity of the hydroxyl group in BD,) what is the evidence that the catalyst enhance the nucleophilicity of the hydroxyl group. The catalyst could coordinate to the carbonyl oxygen of the carbonate increasing the electrophilicity of the carbonyl carbon
LINE 105 (Zn(Acac)2 is 69 400 g/mol, with PDI and yield of 1.64 and 93.7%, respectively) please correct the numbers (delete the space)
LINE 108 (8300, 28 200 and 53 200 g/mol at the same conditions) please correct the numbers (delete the space)
LINE 153 (energy of Zn2P3) should be corrected to Zn 2p3
LINE 155 (Zn(II)[xx]. The binding energy of ZnCl2 appeared at xx) what xx stands for!
I think that the first one is for a missing reference and the second one for a missing value
LINE 185 (atoms with more-hindered in undesirable side reactions[高分子学报].) why this cited reference is written in chinse!
LINE 196 ( time was beyond120 min) check the spaces
LINE 197 (as the reaction time prolonging,) change to as reaction time prolonged
LINE 199 (would rapidity) change to would rapidly
LINE 205 (decrease with prolong of reaction time,) please check the grammar
Author Response
Response to reviewer 3
Dear editor and reviewers,
Thank you very much for your valuable comments on our manuscript “Synthesis of aliphatic polycarbonates from diphenyl carbonate and diols over zinc (II) acetylacetonate” (Manuscript ID: Molecules-2061087) and give me so much useful comments to further perfect our manuscript. We have revised our original manuscript carefully based on the suggestions of reviewers and made the answers according to the reviewer’s questions point by point. The following is the answers to the referee’s points:
Reviewer #3
The authors have explored the Synthesis of aliphatic polycarbonates from diphenyl carbonate and diols over zinc (II) acetylacetonate. Reaction optimization was done, and a little substrate scope has been investigated. The manuscript is poorly set up and the language of the manuscript is poor. consequently, extensive English editing is required.
A: Thanks very much for your so careful review, we feel very sorry to bring you so much trouble for our mistakes. The whole manuscript has been carefully checked and revised to avoid any type mistake, and we also have invited a native English speaker for language polish. We hope that now the language in the revised manuscript can meet with the requirement of Molecules journal. The detailed corrections are highlighted using different colors of fonts in the resubmitted version.
(1)LINE 55 (aliphatic diols to synthesis) should be to synthesize:
A: We were sorry for our careless mistakes, and thanks a lot for your so careful review. This wrong statement of “synthesis” has been corrected as “synthesize”.
(2) LINE 64 (was purification by recrystallization) should be changed to (was purified by)
A: We were sorry for our careless mistakes, and thanks a lot for your so careful review. This wrong statement of “was purification by recrystallization” has been corrected as “was purified by”.
(3) LINE 82 (Mw and PDI of the obtained) there is no definition for PDI throughout the manuscript. Authors should give a definition for its first appearance in the manuscript.
A: The Mw and PDI have been defined as “weight-average molecular weight” and “polymer dispersity index” when they first appearance in the in 2.3. Characterization section.
(4) LINE 100 (DPC with BD to synthesis) should be changed to (to synthesize)
A: We were sorry for our careless mistakes, and thanks a lot for your so careful review. This wrong statement of “synthesis” has been corrected as “synthesize”.
(5) LINE 105 (Zn(Acac)2 is 69 400 g/mol) please correct the numbers (delete the space)
A: The spaces in all the numbers have been deleted in the resubmitted version.
(6) LINE 102, 103 (this reaction scarcely occurred in the absence of a catalyst due to the relatively low nucleophilicity of the hydroxyl group in BD,) what is the evidence that the catalyst enhances the nucleophilicity of the hydroxyl group. The catalyst could coordinate to the carbonyl oxygen of the carbonate increasing the electrophilicity of the carbonyl carbon
A: The transesterification process is a common organism unit reaction, which has been widely investigated in previous works (ChemCatChem,2020,12: 5858-5879). The activation of hydroxyl group over acidic or basic catalysts were thought to be the rate-determining step for transesterification reaction by capturing the proton of hydroxyl group to obtain a high nucleophilicity alcohol anion, which would nucleophilic attack the carbonyl carbon of DMC to initiate this transesterification reaction (Chemical Engineering Science,2022,258:117760). Additionally, the coordination effect of Zn2+ to the carbonyl group in DPC would increase its electrophilicity, which all facilitate the transesterification reaction (Chem. Res Chin Univ,2016, 32: 512-516, ChemCatChem,2020,12: 5858-5879 and Org. Biomol. Chem., 2005,3, 65-72). This process has been further analyzed in depth in the resubmitted version based on the relevant literature.
(7) LINE 105 (Zn(Acac)2 is 69 400 g/mol, with PDI and yield of 1.64 and 93.7%, respectively) please correct the numbers (delete the space)
A: The space has been deleted in the resubmitted version.
(8) LINE 108 (8300, 28 200 and 53 200 g/mol at the same conditions) please correct the numbers (delete the space)
A: The space has been deleted in the resubmitted version.
(9) LINE 153 (energy of Zn2P3) should be corrected to Zn 2p3/2
A: The energy of Zn2P3 has been corrected as “Zn 2p3/2” in the resubmitted version.
(10) LINE 155 (Zn(II)[xx]. The binding energy of ZnCl2 appeared at xx) what xx stands for! I think that the first one is for a missing reference and the second one for a missing value
A: Thanks very much for your so careful comments, and this sentence has been supplemented, and this sentence should be “One can see that the binding energy of Zn2p3/2 in Zn(Acac)2 catalyst appeared at 1021.7 eV, ascribing to the presence of bivalence Zn(II)[22,23]. The binding energy of ZnCl2 appeared at 1022.8 eV”.
(11) LINE 185 (atoms with more-hindered in undesirable side reactions[高分子学报].) why this cited reference is written in chinse!
A: Thanks very much for your so careful comments, this non-English reference has been corrected in the resubmitted revision.
(12) LINE 196 (time was beyond120 min) check the spaces
A: We have checked the spaces of “time was beyond120 min” and a space has been added into this sentence in the resubmitted revision.
(13) LINE 197 (as the reaction time prolonging,) change to as reaction time prolonged
A: the error of “as the reaction time prolonging” has been changed to “as reaction time prolonged” in the resubmitted revision.
(14) LINE 199 (would rapidity) change to would rapidly
A: The error of “would rapidity” has been changed to “would rapidly” in the resubmitted revision.
(15) LINE 205 (decrease with prolong of reaction time,) please check the grammar
A: We have checked the grammar of this sentence, and the sentence of “decrease with prolong of reaction time,” has been changed to “decrease with the prolongation of reaction time,” in the resubmitted revision.
